# The Complex Dysregulations of CD4 T Cell Subtypes in HIV Infection

**DOI:** 10.3390/ijms25147512

**Published:** 2024-07-09

**Authors:** Manlio Tolomeo, Antonio Cascio

**Affiliations:** 1Department of Health Promotion Sciences, Maternal and Infant Care, Internal Medicine and Medical Specialties, University of Palermo, 90127 Palermo, Italy; antonio.cascio03@unipa.it; 2Department of Infectious Diseases, A.O.U.P. Palermo, 90127 Palermo, Italy

**Keywords:** AIDS, HIV infection, CD4+ T cells subtypes, HAART, resting cells, “Kick and Kill”

## Abstract

Human immunodeficiency virus (HIV) infection remains an important global public health problem. About 40 million people are infected with HIV, and this infection caused about 630,000 deaths in 2022. The hallmark of HIV infection is the depletion of CD4+ T helper lymphocytes (Th cells). There are at least seven different Th subtypes, and not all are the main targets of HIV. Moreover, the effect of the virus in a specific subtype can be completely different from that of the others. Although the most compromised Th subtype in HIV infection is Th17, HIV can induce important dysregulations in other subtypes, such as follicular Th (Tfh) cells and regulatory Th cells (Treg cells or Tregs). Several studies have shown that HIV can induce an increase in the immunosuppressive activity of Tregs without causing a significant reduction in their numbers, at least in the early phase of infection. The increased activity of this Th subtype seems to play an important role in determining the immunodeficiency status of HIV-infected patients, and Tregs may represent a new target for innovative anti-HIV therapies, including the so-called “Kick and Kill” therapeutic method whose goal is the complete elimination of the virus and the healing of HIV infection. In this review, we report the most important findings on the effects of HIV on different CD4+ T cell subtypes, the molecular mechanisms by which the virus impairs the functions of these cells, and the implications for new anti-HIV therapeutic strategies.

## 1. Introduction

HIV infection is characterized by a variety of immunological abnormalities [1]. The most important is the decrease in the number and function of CD4+ T helper lymphocytes (Th cells). Although these cells play an important role in the immune defense against HIV, they are the same cells that HIV targets and kills during the infection, causing a state of immunodeficiency that results in the onset of opportunistic infections, such as fungal infections, toxoplasmosis, tuberculosis, and some tumors, such as Kaposi sarcoma and lymphomas.

Currently, highly active antiretroviral therapy (HAART) based on different combinations of anti-HIV specific drugs, such as nucleoside and non-nucleoside reverse transcriptase inhibitors (NRTIs and NNRTIs), protease inhibitors (PIs), and integrase inhibitors (II), can control HIV and prevent complications. However, a complete cure is not possible because some latently infected Th cells (resting cells) form a reservoir of the virus. In these cells, antiretroviral drugs are ineffective, and an interruption of HIV treatment causes the reemergence of viremia [2,3].

Mature effector Th cells originate from naïve CD4+ T helper lymphocytes that are produced in the thymus. Here, a cell selection takes place, and only self-tolerant naïve CD4+ T helper cells are released into the blood. These cells remain in a non-proliferating (resting) state until they encounter an antigen presented by specific antigen-presenting cells (APCs). This induces the activation, proliferation, and differentiation of immature naïve CD4+ T helper cells that become mature effector Th cells. Some effector cells can return to a non-proliferating resting state, becoming memory Th cells characterized by a long half-life. Memory cells can be classified into central memory T cells (TCM), where HIV seems to prevalently persist, and effector memory T cells (TEM). TCM cells are mainly located in lymphoid organs and blood, while TEM cells are mainly in blood, but unlike TCM, can circulate through peripheral organs. If these memory T cells are derived from HIV-infected effector cells, they form a primary reservoir for latent HIV proviruses. This reservoir can also be produced by the direct infection of memory Th cells [4].

In memory resting Th cells, the proviral DNA is integrated into their chromosomes but there is no production of the virus. Cell cycle progression of HIV-infected cells is required for efficient RNA reverse transcription, proviral DNA integration, and virus production [5,6]. Currently, available anti-HIV drugs act only on specific stages of viral replication, so they are ineffective in quiescent infected cells, such as resting cells, where the virus is not in the replicative phase [7,8,9].

The role of resting cells and the impossibility of healing HIV infection have been extensively studied in recent years. New insights have been gained by studying the effects of the virus on different types of Th effectors. There are different subsets of Th effector cells and not all of them are the main targets of HIV. Seven subsets of Th effector cells are currently known and well studied, namely Th1, Th2, Th9, Th17, Th22, Tfh, and Tregs. The differentiation in each subtype depends on the cytokine environment present during antigen recognition (Figure 1).

All Th subtypes are potentially susceptible to HIV infection, but the infectious capacity of the virus can differ depending on the type of Th effector cell. (Table 1). This mainly depends on the strain of HIV and the expression of coreceptors in the surface cell membrane of infected cells. To infect Th cells, the HIV envelope protein, gp120, must interact with the CD4 receptor and the CCR5 or CXCR4 coreceptor. Two types of HIV strains exist: (1) the R5 strain that enters and infects host CD4 cells by attaching to the CCR5 coreceptor; (2) the X4 strain that infects CD4 cells by attaching to the CXCR4 coreceptor. HIV is usually R5-tropic (uses CCR5) during the early stages of infection, but the virus may later switch to using either only CXCR4 (X4-tropic) or both CCR5 and CXCR4 (dual-tropic).

Although Th1 and Th2 immune responses, the first to be studied, play important functions in protecting our body from intracellular and extracellular infections, HIV seems to attack mainly other Th effector cells, such as Th cells implicated in the defense against opportunistic infections (Th17), in the regulation of cellular immunity (Tregs), in the activation of B cells for the production of antibodies (Tfh), and the protection of gut epithelium (Th22) (Table 1). All these Th subtypes can mature in memory T cells. Th1, Th2, Th17, Th22, and Treg cells mature into effector memory T cells, while Tfh and Treg cells can mature into central memory T cells [10,11].

Most HIV-infected Th cells are located within lymphoid tissue of the gastrointestinal tract (gut-associated lymphoid tissue or GALT) and the lamina propria of the gut [12]. Th cells residing in these areas are primarily infected and depleted. The intestinal barrier, which must limit the entry of microbes from the gut into the systemic circulation, can be destroyed by HIV infection and the increased intestinal permeability leads to the translocation of microbial products from the gut to the blood, contributing to increased immune activation, risk of non-AIDS-related comorbidities and mortality in people living with HIV [12,13]. However, reviewing data from different mammalian species, other authors have found that only 5%–20% of all lymphocytes reside in the gut, suggesting that the spleen and lymph nodes, rather than the gut, are the largest immune compartments in mammals [14].

Th17, Treg, and Th22 cells are the main targets of the virus and are depleted (Table 1). In contrast, Tfh cells can be infected, but rarely they are depleted (in some cases they are increased) [15]. However, their function is greatly impaired by the virus, and this causes deregulation in B cell function and production of protective anti-HIV antibodies (Table 1).

**Table 1 ijms-25-07512-t001:** Susceptibility of Th subtypes to HIV infection, and effects caused by their infection.

Th Cell Subtype	Susceptibility to HIV Infection	Effects of HIV Infection	References
Th1	Susceptibility for R5 and X4 strains. Moderate levels of integrated HIV DNA	Th1 cell depletion	[16,17]
Th2	High for X4 strain	Th2 cell depletion	[18]
Th9	Very high for X4 strain	Th9 cell depletion	[19]
Th22	High for R5 strain	Th22 cell depletion. Stromal cell depletion and impairment of intestinal epithelial integrity	[20,21]
Tfh	High for R5 strain	No or scarce cell depletion, but impairment of their functions and humoral immunity	[22,23,24,25,26,27,28,29,30,31,32,33,34]
Th17	Very high for R5 and X4 strains	Strong cell depletion and severe impairment of intestinal epithelial integrity. Emergence of opportunistic infections	[35,36,37,38,39,40,41,42,43,44,45,46,47,48]
Tregs	Very high for R5 and X4 strains	No Treg cell depletion in the early phase of infection. Increased immunosuppressive activity of Tregs	[49,50,51,52,53,54,55,56,57,58,59,60,61,62,63,64,65,66]

## 2. The Effects of HIV on Th1, Th2, and Th9 Cells, the Th1/Th2 Shift

The discovery of two different mouse helper T cell clones, defined as Th1 and Th2, was first reported in 1986 [67]. They were the first two subtypes of Th cells to be discovered and play a very important role in the defense against intracellular microorganisms (Th1) and in the production of antibodies (Th2). Recently, a Th subtype similar to Th2, defined as Th9, has been identified. These Th subtypes are infected by HIV, which induces their death by apoptosis. However, other cell subtypes appear to have a more important role in HIV infection and evolution in AIDS, the most severe stage of HIV infection.

### 2.1. Th1 Cells

Based on transcription factors and cytokine production, in 1986, Mossman et al. delineated two main classes of helper cells: Th1 and Th2. Th1 cells promote cell-mediated immune responses required for defense against intracellular pathogens through the production and secretion of cytokines such as IFN-gamma, interleukin (IL)-2, and TNF-alpha/beta. These cytokines promote macrophage activation, nitric oxide production, and cytotoxic T lymphocyte activation and proliferation, leading to the destruction of intracellular pathogens. Therefore, the main role of Th1 cells is the regulation of macrophage and monocyte activities. Th1 differentiation is induced by IL-12 which is secreted by antigen-presenting cells (APCs). IL-12 activates STAT4 in naïve CD4+ T helper cells, which induces the differentiation in Th1 cells [67,68,69]. In addition, IL-12 induces the expression of T-bet (T-box expressed in T cells), which is implicated in the transcription of Th1-specific genes [70]. Gosselin et al. observed that CXCR3+CCR6− T cells (Th1) expressed CCR5 and CXCR4 but were relatively resistant to R5 and X4 HIV in vitro and CCR6+ T cells (such as Th17 and Th1Th17) compared with CCR6− T cells (such as Th1) and harbored higher levels of integrated HIV DNA [16]. Singh et al. reported that HIV-1 infection resulted in a greater loss of HIV-1 specific Th17 cells than the HIV-1 specific Th1 cells, suggesting a higher susceptibility of the Th17 cells to the infection. Although the gradual loss of Th1 cells was also observed during HIV-1 disease progression, Th1 cells were found to be more resistant to HIV-1 infection compared to Th17 [17]. A possible explanation for the increased resistance of Th1 cells to HIV infection can be raised by considering the work of Galio et al. [71]. T-bet, which is the Th1 master transcription factor, antagonizes GATA-3, a factor that plays a central role in regulating Th1 and Th2 cell differentiation. GATA-3 can bind to the promoter region of the HIV genome (designated as long terminal repeat or LTR), inducing HIV transcription [71]. Therefore, T-bet may potentially limit HIV transcription.

### 2.2. Th2 Cells

Th2 cells promote the activation of the humoral (antibody-mediated) immune response against extracellular parasites through the production of specific cytokines such as IL-4, IL-5, IL-6, IL-9, IL-13, and IL-25. These cytokines induce strong antibody production and recruit eosinophils and basophils to the site of infection or in response to allergens or toxins. Th2 cells also control the regulation of B cell class-switching to IgE. Th2 differentiation is induced by IL-4, which causes the activation of STAT6 in naïve CD4+ T helper cells, which promotes the Th2 response and the production of Th2-specific cytokines [18]. Th2 cells are mainly susceptible to HIV X4 replication because CCR5 is expressed at undetectable levels on Th2 cells, while CXCR4 is highly expressed in their cell surface membrane.

### 2.3. Th9 Cells

Th9 cell is a newer subset of CD4 T cells discovered in 2008 [72]. Like the Th2 subtype, Th9 cells differentiate from naïve CD4+ T helper cells in the presence of IL-4. However, to differentiate into Th9, naïve CD4+ T helper cells also need transforming growth factor β (TGF-β) [73]. IL-4 activates STAT6, while TGF-β activates the transcriptional factors Smad2 and Smad4 that are required for Th9 differentiation [74]. Th9 cells play a main role in the defense against helminthic infections. IL-9 promotes the survival and proliferation of anti-parasitic leukocytes, including mast cells, eosinophils, and basophils, and increases IgE and IgG generation from B cells. Of interest, Th9 cells have been shown to possess greater anti-tumor activity than other subgroups of Th [75,76]. Like Th2 cells, Th9 are poorly susceptible to R5-tropic HIV infection. Interestingly, the susceptibility to the X4 strain infection is more pronounced in Th9 than in Th2 cells. Th9 cells are a long-lived CD4+ T cell population, but it is unclear whether they represent a specific viral reservoir for the X4 HIV strain [19].

### 2.4. The Th1/Th2 Shift

Several authors have reported that a switch from Th1 to Th2 cytokine production occurs during HIV infection. Clerici and Shearer observed that the progression of HIV infection in AIDS is characterized by the loss of IL-2 and IFN-γ production and the increase in IL-4 and IL-10. They proposed that the shift from Th1 to Th2 cytokine production contributes to the progression of AIDS in HIV-infected patients [77]. Klein et al. observed, through cytoplasmic cytokine detection on a single cell level by flow cytometry, a continuous decrease in IFN-γ production in CD4+ T cells during HIV infection and a marked reduction in IL-2 expression in CD4+ T cells in patients with AIDS. In contrast, CD4+ T cells expressing Th2 cytokines increased during HIV infection. The maximum frequency of CD4+ T cells expressing IL-4 was seen in HIV-infected individuals without AIDS, whereas the rate of IL-10-producing CD4+ T cells was highest in patients with AIDS [78]. Becker proposed that the shift from Th1 to Th2 cytokine production during HIV-1 infection is indicative of an allergic response to viral proteins that may be reversed by Th2 cytokine inhibitors [79]. However, not all authors have observed a Th1-Th2 shift in HIV-infected patients, and there is no unanimity of opinion about the role of Th1 cell depletion in determining some opportunistic diseases [80,81,82,83]. However, people living with HIV have up to a 20 times higher risk of developing active tuberculosis compared to those without HIV infection, and in high-prevalence countries, it has been reported that one of the leading causes of HIV-related death is co-infection with tuberculosis, which is believed to be largely a target of Th1 immunity. Genetic defects in the IL-12/IFN-γ axis are associated with disseminated infections. Some reports suggest that the control of tuberculosis infection may be best correlated with CXCR3+CCR6+ Th1Th17 cells, a Th subtype enriched with integrated HIV DNA [84,85]. These cells co-express retinoic acid-related orphan receptor gamma t (RORγt), like Th17 cells, and T-bet and produce both interleukin IL-17 and interferon IFN-γ [86]. The Th1Th17 subtype appears to play a role in the early stages of tuberculosis because it has been documented in pulmonary tuberculosis outbreaks in patients with recent tuberculosis [87]. Several pieces of evidence indicate that *mycobacterium tuberculosis* and HIV mutually support each other during coinfection by multiple mechanisms. *Mycobacterium tuberculosis* secretes EspR, a transcription regulator protein that can bind to the promoter of the host IL-4 gene, inducing the production of high levels of IL-4 that induce Th2 cell differentiation, shifting the macrophage polarization to an M2 state that impairs the clearance of the intracellular *mycobacteria* [88].

Therefore, although there is no unanimity of opinion about the Th1/Th2 shift in HIV infection, increased production of Th2 cytokines may occur under specific conditions in HIV-infected individuals and can play a role in the emergence of specific infections such as tuberculosis.

## 3. Th22: The Defender of the Intestinal Mucosa

Th22 is a new, independent, and stable subset of Th cells that express high levels of the CCR5 coreceptor. Although they are sensitive to R5 strain entry, they may have protective effects against HIV infection [89]. The main effector molecule secreted by these cells is IL-22. They are also able to secrete IL-26, IL-13, TNF-α, and granzyme B [90]. Th22 cells express the skin chemokine receptors CCR4, CCR6, and CCR10. This explains why they are prevalently found in the skin and fewer found in blood circulation. Th22 differentiation is mainly induced by IL-6 and TNF-α. However, IL-21 and IL-23 can also induce the differentiation of Th22 cells [91,92]. These interleukins activate specific transcription factors such as STAT3, RORγt, and AhR that regulate Th22 differentiation. However, AhR seems to be the main regulator of Th22 differentiation. Keratinocytes can secrete IL-23, which binds to the IL-23 receptor of skin DCs, inducing the secretion of IL-6 and TNF-α. This induces the differentiation of naïve CD4+ T helper cells into Th22 cells in the skin [93]. Considering that HIV infection is initially characterized by a structural deterioration of the gut epithelium that causes impaired epithelial barrier function and increased levels of microbial translocation, Th22 cells may have a central role in the first phases of HIV infection. Th22 cells are highly sensitive to HIV infection because they express high levels of coreceptors. Like Th17 cells, Th22 cells are drastically depleted in the intestinal epithelium during HIV infection. Since IL-22 has epithelial reparative and regenerative properties, depletion of this cytokine in the gut causes impairment of epithelial integrity and microbial translocation. In contrast, high levels of IL-22 were observed in some people resistant to HIV infection (individuals exposed to HIV but uninfected), indicating that Th22 cells may play a protective role in HIV infection. Oliviera et al. showed that HIV-exposed uninfected individuals exhibit an increased frequency of CD4+ IL-22 secreting T cells, whereas HIV-infected partners showed a high frequency of CD4+ IL-17+ T cells in response to p24 HIV protein [20]. The protective effect of Th22 seems correlated to the ability of IL-22 to produce acute-phase proteins such as the acute-phase serum amyloid A (A-SAA). This protein, which is upregulated in repeatedly HIV-exposed uninfected individuals, induces the phosphorylation of CCR5, downregulation of CCR5 expression, and reduced susceptibility of cells to HIV infection [21]. The reason why some individuals have high IL-22 production and are therefore protected from HIV infection is not known, but it suggests that Th22 cells could be a target for future approaches in HIV treatment.

## 4. HIV Infects Tfh Cells, Impairing Humoral Immunity

Tfh is a subset of Th cells located in the B cell follicles of secondary lymphoid tissues that play a central role in supporting B cell activation, high-affinity antibody production, antibody class switching, and memory B cell production after infection [94]. Tfh cells express the chemokine receptor CXCR5 (chemokine receptor C-X-C type 5) on their surface membrane, which allows Tfh cells to migrate into the B cell areas of the lymph node and the receptor PD-1 (program death-1). Although the PD-1 receptor is associated with negative regulation of immune response and functional exhaustion of T cells, it is required in Tfh cells for optimal germinal center (GC) localization, IL-21 production, and B cell affinity maturation [95]. Tfh cell differentiation is a multi-step process that begins in the T cell zone of lymph nodes through the interaction between the receptor ICOS (inducible T cell costimulator receptor) of naïve CD4+ T helper cells and ICOS-L of DCs. This interaction induces the expression of CXCR5 on the surface membrane of naïve CD4+ T helper cells that become pre-Tfh cells. In this early phase of Tfh differentiation, DCs secrete IL-6 and IL-12, which induce the production of Bcl-6 in a STAT3-dependent manner in pre-Tfh cells [96,97]. Pre-Tfh cells migrate to the TB border zone of the lymph node, where they interact with B cells, completing their maturation. Moreover, in the T-B border zone of the lymph node, the interaction between Tfh cells and B cells causes further production of Bcl-6. Finally, Tfh cells migrate to the GC, where they provide a further signal to the B cells. Bcl-6 is a transcription factor that activates several genes critical to Tfh function and inhibits other genes for non-Tfh cell differentiation [98]. It is co-expressed with the transcription factor c-Maf, which contributes to inducing other important functions in Tfh cells, such as IL-21 and IL-4 production [99]. Tfh cells provide different cytokines that are able to induce the activation of B cells. IL-21 and IL-4 produced by Tfh cells are required for plasma cell differentiation and somatic hypermutation, while IL-9 plays a key role in memory B cell formation in the GC.

Although Tfh cells do not express CCR5, they are infected by HIV. Unlike mature Tfh cells, Tfh precursors (pre-Tfh cells) express the coreceptor CCR5. Therefore, the infection of this CD4 T cell subset may occur in the early phase of Tfh differentiation [22,23]. Furthermore, there is a subset of CD4 central memory T cells in peripheral blood termed peripheral Tfh (pTfh) cells, expressing CXCR5 and similar in function to Tfh cells. Different from Tfh cells that interact with B cells within GC, Tph cells provide help to memory B cells within inflamed tissues. Although they do not express high levels of CCR5 coreceptors on their surface membrane, these cells are highly susceptible to HIV infection. Of note, the virus can also persist in these cells after plasma virus suppression with potent antiretroviral therapy, indicating that they can represent an important reservoir of the virus, contributing to HIV latency [24]. Banga et al. showed that CD4 T cells in lymph nodes (LN CD4 T cells) that express PD-1, which are composed of about 65% Tfh cells as defined by the expression of the cell surface receptors CXCR5 and PD-1, are the major source of replication-competent HIV-1 and of the infectious virus, as compared to any other (CXCR5-PD-1- and CXCR5+PD-1-) blood or lymph node memory CD4 T cell populations. Therefore, these cells may represent an obstacle to finding a functional cure for HIV-1 infection [25].

The number of Tfh cells in HIV-infected patients is not particularly reduced (indeed, in some cases, it is increased) [15], but their function is greatly impaired. HIV-infected individuals show significant deregulation in B cell function and fail to produce protective anti-HIV antibodies. Despite the occurrence of hypergammaglobulinemia in HIV infection, specific antibody production and in vitro B-cell differentiation responses are frequently impaired. Signs of aberrant B cell hyperactivity, including hypergammaglobulinemia [26], spontaneous secretion of immunoglobulins in culture [27], and increased expression of activation markers [28], have been observed in HIV-infected individuals. However, paradoxically, HIV-infected patients have an impaired humoral response [29], and their B cells respond abnormally when stimulated ex vivo [30]. Chirmule et al. observed that B cell perturbation in HIV-infected patients can be caused by impaired GC T-helper cell function for B cells [31]. Infected Tfh cells are unable to provide effective signals to B cells in the GC and this results in an inefficient anti-HIV antibody response and premature death of memory B cells. Moir et al. observed that viremia was also associated with the appearance of a subpopulation of B cells that expressed reduced levels of CD21. These B cells dramatically reduced proliferation in response to B cell stimuli and enhanced secretion of immunoglobulins when compared with normal B cells. The presence of this B cell subpopulation that fails to proliferate in response to B cell stimuli and secrete high levels of immunoglobulins may explain the hypergammaglobulinemia associated with HIV disease [32]. Cubas et al. observed a higher frequency of PD-L1 positive B cells in GCs of HIV-infected individuals, leading to excessive and persistent triggering of PD-1 on Tfh cells. This persistent triggering affected the capacity of Tfh to provide B cell help by decreasing the expression of ICOS that, in turn, causes decreased transcription of c-Maf and subsequently reduced secretion of IL-21 and IL-4 [33]. In addition, Chakhtoura et al. demonstrated that the transcriptional profiles of HIV-positive and HIV-negative proliferating GC-Tfh cells were different, indicating that the virus can alter the transcriptome of proliferating GC-Tfh cells, resulting in a loss of their ability to provide adequate help to GC-B cells. In fact, HIV is able to target a large number of genes in Tfh cells that are critical to the interaction with GC-B cells. In particular, targeting the gene encoding cMAF, HIV alters cMaf signaling and Tfh functions [34]. The key role of Tfh cells in antibody production and the dysfunction observed in Tfh cells of HIV-infected individuals provide a possible explanation for the difficulty of obtaining effective HIV vaccines capable of inducing a protective antibody response.

## 5. Th17 and Treg Cells: “Dr. Jekyll and Mr. Hyde” of Immunity in HIV Infection

Th17 cells and Treg cells are two of the main targets of HIV, and their depletion (Th17) and dysregulation (Tregs) contribute to the evolution into AIDS [35,36,37,38,39,40,41,100]. There is a complex relationship between Th17 cells and Tregs in HIV infection. Th17 cells play a crucial role in the defense and prevention of opportunistic infections. The drastic reduction in Th17 cells during HIV infection leads to the appearance of opportunistic infections that are the main cause of death in patients affected by AIDS [39,40,41]. Therefore, preservation of Th17 numbers and functions is the primary goal of HIV treatment. Tregs are a subset of CD4+ T cells with immunosuppressive functions. Recent studies of well-controlled HIV infection in patients on antiretroviral therapy have shown higher frequencies of inducible, intact proviruses in Tregs compared to other CD4+ T cells [100]. In other studies, the total HIV DNA burden associated with Tregs was similar to that observed in memory T cells [101]. However, the number of Tregs in HIV-infected patients is often normal or declines very slowly while their immunosuppressive activity is greatly increased. This contributes to aggravating the immunosuppressive status of patients. Treg and Th17 cells have the potential to convert to each other depending on the cytokine environment. This conversion was first reported in studies on mice, where IL-6 was able to convert Tregs in Th17 cells in the absence of TGF-β [102,103]. In humans, Tregs can differentiate into Th17 depending on the nature of the stimulus provided [104,105]. In contrast, the products of the catabolism of tryptophan, which is increased in HIV-infected patients, can activate Tregs and block their conversion into Th17 cells. Just as the phrase “Jekyll and Hyde” is used in popular culture to refer to a person with good but sometimes surprisingly evil natures, after the great success of Robert Louis Stevenson’s novel “The Strange Case of Dr. Jekyll and Mr. Hyde” so Th17 and Treg cells can be considered two sides of the same coin, but with opposite effects for the immune system and for the prognosis of HIV infection. This has led several authors to consider Tregs as a potential target for HIV immunotherapy to be used in combination with HAART.

### 5.1. Th17 Cells

Th17 cells are a subset of T helper cells with pro-inflammatory activity characterized by the expression of the nuclear receptor transcription factor RORγt (retinoic acid-receptor (RAR)-related orphan receptor transcription factor) encoded by the gene *RORC2* [106,107,108,109]. Th17 cells are located primarily in the gastrointestinal tract [110,111] and differentiate from naïve T cells under the synergistic action of TGF-β and IL-6 [112]. These cytokines are produced by APCs after contact with pathogens. TGF-β promotes Th17 cell differentiation by upregulating the expression level of the IL-23 receptor (IL-23R). IL-6 and IL-23 activate STAT3, which induces the expression of Th17 cell-specific transcription factor RORγt. This transcription factor promotes the production of Th17 specific factors, including IL-17a, IL-17F, IL-22, IL-26, IL-23R, CSF-2, CCR6, and CCL20 [113]. IL-17A and IL-17F bind to a heteromeric receptor complex sited in the surface membrane of different cells, such as neutrophils, monocytes, fibroblasts, and endothelial cells. In endothelial cells, IL-17 induces a specific pattern of chemokines, such as CXCL1, CXCL2, and CXCL5, that sustains neutrophil cell influx to sites of inflammation [114]. In addition, Th17 cells produce granulocyte–macrophage colony-stimulating factor (GM-CSF), which supports the proliferation and differentiation of neutrophils [115] and are able to induce B cell recruitment through CXCL13, which is a potent chemotactic factor for B cells [116]. The main functions of Th17 cells are the defense of gut mucosal against microbial pathogens [117], and the prevention of opportunistic infections, including fungal infections (*Candida albicans*, *Cryptococcus neoformans*), *Toxoplasma gondii* [118], and *Pneumocystis jirovecii* [119].

Th17 cells express high levels of CD4, CCR5, and CXCR4 and are extremely susceptible to HIV infection, showing higher levels of the integration of HIV DNA compared to Th1 and Th2 cells [16,38,42]. The high production of the virus after infection and the high levels of the DNA integration of HIV into Th17 cells indicate the presence of post-entry mechanisms that enhance viral replication. It has been observed that Th17 cells have increased NF-κB (nuclear factor-kappa B) nuclear translocation and DNA binding activity compared to Th1 cells [43]. NF-κB is able to bind to the promoter of the HIV enhancer region LTR, which contains two NF-κB binding sites, supporting HIV genome transcription [44]. Finally, Th17 cells show a low expression of RNase A, a superfamily protein capable of inhibiting HIV production [38].

During HIV infection, Th17 cells are depleted from the intestinal mucosa, genital mucosa, and blood. HAART can only partially restore the number of Th17. A complete recovery of these CD4+ cell subsets can be reached only when HAART is started in the acute phase of HIV infection but not when initiated during the chronic phase [45]. The drastic reduction in Th17 cells during HIV infection leads to the appearance of opportunistic infections and an important impairment of the antimicrobial immunity in the intestinal mucosa, which results in an increase in the blood of microbial markers [46]. AIDS, the most severe stage of HIV infection, is characterized by the manifestation of opportunistic infections such as bacterial infections (including tuberculosis, *salmonella* infection, and the *Mycobacterium avium complex* infection), viral infections (such as *cytomegalovirus* and *herpes simplex* virus 1 infections), fungal infections (like yeast infections, *pneumocystis jirovecii* infection, and histoplasmosis), and parasitic infections (such as cryptosporidiosis and toxoplasmosis). The translocation of bacteria and fungi in the blood induces chronic systemic immune activation and tissue inflammation which can be the cause of cardiovascular disease, neurocognitive impairment, and loss of quality of life. Recently, it has been proposed that Th17 cells may represent an important viral reservoir for HIV, not only because of their high susceptibility to infection but also because they have properties similar to stem cells [47]. Th17 cells are long-lived, exhibit self-renewal properties, and can enter the memory pool with high efficiency. The self-renewal capacity observed in Th17 cells appears to depend on the high expression of Tcf7 (Transcription Factor 7), a protein that confers self-renewing properties. These properties and the observation that HIV DNA levels in Th17 cells remain stable over many years of antiretroviral therapy indicate that these cells may represent a reservoir for latent HIV proviruses [48].

Th1/Th17 cells are a subtype of Th cells that co-express the chemokine receptors CCR6 and CXCR3 and the Th17 and Th1 lineage-specifying transcription factors RORγt and T-bet [120]. They are characterized by the ability to co-produce IL-17 and IFN-γ and to undergo self-renewal and homeostatic proliferation [47]. Th1Th17 cells express high levels of CCR5 and are more susceptible to HIV infection compared to CXCR3+ Th1 cells [85]. In addition, central memory CD4+ T-cells that co-expressed CXCR3 and CCR6 in HIV-infected individuals receiving antiretroviral therapy expressed high levels of integrated HIV DNA [85]. The self-renewal property, together with the high expression of integrated HIV DNA, makes these cells a potentially important reservoir for the virus [85].

### 5.2. Treg Cells

Treg cells are a subset of CD4+ T cells with immunosuppressive functions that maintain tolerance to self-antigens and prevent autoimmune diseases. They express CD4, low levels of CD127, high levels of CD25 and CTLA-4 on the cell surface membrane, and the transcription factor FoxP3 (forkhead box P3), whose function is crucial for the immunosuppressive activity of these cells [121,122,123]. Furthermore, they can express PD-1. Naïve CD4+ T cells differentiate into Tregs in the presence of IL-2 and TGF-β. Treg cells are mainly found in the intestine, where they induce tolerance to exogenous antigens. The most important cytokines with inhibitory effects on the immune system produced by these cells are TGF-β, IL-35, and IL-10 [124]. In addition, they produce granzyme B and perforin, which can induce apoptosis of other immune effector cells [125,126,127,128].

Tregs can be classified into thymic Tregs (tTregs), which are differentiated in the thymus from T cells produced in bone marrow, and peripheral Tregs (pTregs), which are derived from circulating naïve CD4+ T helper cells in the presence of IL-2 and TGF-β. tTregs express higher levels of CD25 and CTLA-4 (cytotoxic T lymphocyte antigen 4) and are the majority of Treg cells in the body. Like FoxP3, CTLA-4 is critical to the effective suppression of the immune system, and a deficiency in this factor impairs the suppressive function of Tregs. CTLA-4 interacts with CD80 (differentiation cluster 80) and CD86 receptors in DCs and B cells [129]. These two receptors are also the ligands of the CD28, which is a protein expressed on T cell membrane surfaces. The interaction between CD28 and CD80 or CD86 provides costimulatory signals required for T cell activation and survival. In contrast, the interaction of CD80 or CD86 with CTLA-4 inhibits T-cell effector functions. Moreover, CTLA-4 expressed on Tregs is able to reduce the expression of CD80 and CD86 receptors from the cell surface of DCs through a process defined as transendocytosis [130]. Chevalier et al. observed that the percentage of Tregs and their subsets were similar in HIV-infected individuals and healthy subjects. However, Tregs from HIV-infected patients showed a higher expression of FoxP3 and CTLA-4 than Tregs from noninfected individuals, suggesting that infected Tregs may have a greater immunosuppressive activity [49]. In addition, Xiao et al. observed that the proportion of Tregs expressing PD-1 in HIV-1-infected individuals was significantly higher than that observed in noninfected subjects. PD-1 confers important suppressive properties on Tregs. In fact, PD-1 binds to the PD-1 ligand on the surface membrane of T cells and induces apoptosis of T cells and immunosuppression. The high proportion of PD-1+Tregs in HIV-infected patients is associated with disease progression, and their levels are positively correlated with viral load and exhaustion of CD4 and CD8 T cells [50]. In the intestinal mucosa, the increased activity of Tregs and the reduced number of Th17 cause a deficit in the local immune activity that leads to microbial translocation into the blood and systemic immune activation [51]. Furthermore, Tregs promote collagen deposition and fibrosis in the intestinal mucosa and mesenteric lymph nodes through the secretion of TGF-β1 [52]. Lymph node fibrosis limits interactions between B- and T cells, restricts the trafficking of resident cells, and decreases their activation and proliferation [53]. Although antiretroviral therapy can normalize the number of Tregs in the blood, it cannot reverse the imbalance between Th17 cells and Tregs in GALT and the damage in the intestinal mucosa [54,55]. Interestingly, the increased immunosuppressive activity of Tregs, the imbalance between Treg and Th17 cells in GALT, and the fibrosis of intestinal mucosa and mesenteric lymph nodes has also been observed in long-term HIV non-progressives [55] and elite controls who are HIV infected individuals who spontaneously control viral replication [56]. In addition, antiretroviral therapy fails to reduce the expression of CTLA4 and PD-1 in Tregs.

The CD39-adenosine pathway is another mechanism used by Tregs to inhibit an HIV-specific immune response. Adenosine is a metabolite that regulates several physiological processes and is produced by ATP, which is de-phosphorylated to adenosine by the ectonucleotidases CD39 and CD73 [57]. Adenosine inhibits the activity of several immune cells after binding with its specific receptor (A2AR), which is expressed in different cells of the immune system, including T cells. A2AR activation inhibits the secretion of neutrophil chemokines and increases protein kinase A (PKA) activity that causes the inhibition of MAP kinases, protein kinase C, NF-κB, and NF-AT (activated T nuclear factor) in T cells. In addition, the activation of the A2AR signal on Tregs induces an increase in PD-1 and CTLA-4 [58]. Nikolova et al. observed that HIV-infected patients exhibit a higher frequency of Treg CD39+ cells and that Tregs CD39+ correlates inversely with CD4 T cell counts. Moreover, T cells from these patients showed a higher expression of A2AR and higher in vitro sensitivity to the suppressive effect of adenosine [59].

BACH2 and STAT5B are two other transcription factors that play important roles in the development and function of Tregs. In CD4+ T cells, HIV can integrate its genome within the transcription unit of expressed genes, producing chimeric transcripts harboring HIV sequences fused to cellular exon sequences [60,61]. Cesana et al. observed that *BACH2* and *STAT5B* are two genes most frequently targeted by HIV and that the promoter region of the HIV genome (LTR) could directly control the expression of these genes, driving the formation of large amounts of chimeric mRNA transcripts that encode for unaltered full-length wild-type STAT5B and BACH2 proteins. This was observed in Tregs and in central memory T cells that are resting cells harboring HIV [62]. BACH2 and STAT5B are transcription factors that play important roles in the development and function of Tregs. STAT5B is the signal transducer of IL-2, which is required for the activation and proliferation of Tregs and for the expression of FoxP3 [63,64]. BACH2 promotes the development and survival of Tregs, preventing their terminal differentiation and exhaustion [65] and inducing the generation of long-lived memory Tregs. The expression of BACH2 in Tregs makes these cells important viral reservoirs, considering that they contain high amounts of HIV DNA and that long-lived Treg cells harboring HIV insertions can escape from the lytic activity of HIV-specific cytotoxic T lymphocytes due to their suppressive capacity, whereas infected memory CD4+ T cells, which are also important reservoirs for HIV, cannot escape from this lytic activity [66].

### 5.3. The Th17/Treg Ratio

Although there is a correlation between Th17 and Treg cells being able to convert to each other, we do not know why HIV causes a marked reduction in Th17 cells without affecting the number of Tregs or only causing a slow reduction in their number. It is known that the balance between TGF-β, IL-6, and retinoic acid determines the development of Treg or Th17 cells by competition between Foxp3 and RORγt [102]. TGF-β promotes the expression of Foxp3 and RORγt, but Foxp3 is capable of inhibiting RORγt expression. Therefore, when the concentration of TGF-β is very high, the high expression of Foxp3 inhibits the expression of RORγt, thereby inhibiting the differentiation of Th17 cells. Retinoic acid inhibits Th17 polarization and enhances FoxP3 expression, favouring the differentiation of Treg cells. In contrast, the expression of Foxp3 is inhibited by IL-6, and the presence of TGF-β and IL-6 induces the differentiation into Th17 [112]. It is possible to assume that the increase in immunosuppressive capacity of Tregs, without an important reduction in their numbers, could be a strategy of the virus to survive in these cellular reservoirs. Thus, the ratio between Th17 and Tregs is important for understanding the stage and development of HIV infection. In healthy individuals, this ratio is normally between 1.0 and 1.2, while in HIV-infected patients, this ratio ranges from 0.75 to 0.2 [111]. The enzyme indoleamine 2,3-dioxygenase (IDO) may play a role in determining the altered Th17/Treg ratio in HIV-infected patients. IDO catalyzes tryptophan catabolism, and the products of this catabolism, such as kynurenine (Kyn) and 3-hydroxy anthranilic acid (3-HAA), have immunosuppressive activity, being able to activate Treg cells and block their conversion into Th17 cells. Of interest, HIV-infected patients have increased tryptophan catabolism, reduced blood tryptophan levels, and increased blood catabolites of this amino acid, such as Kyn and 3-HAA, compared to uninfected individuals [131,132]. Kyn can induce FoxP3 expression and the generation of Tregs and can inhibit the production of Th17 cells and the expression of RORγt. The proximal catabolite of tryptophan catabolism, 3-HAA, can also lead to a decrease in the Th17/Treg ratio, but the mechanism of action of this catabolite is not yet known [133]. Recently, Yero et al. characterized Treg cell subsets in 103 individuals by flow cytometry, including untreated HIV-infected participants in acute and chronic phases, HAART-treated in early infection, elite controllers, immunological controllers (ICs), and HIV-uninfected controls. They observed that early HAART initiation was unable to control the higher levels of activation and immunosuppressive markers of Treg subsets, even in elite controllers. Moreover, they evaluated the epigenetic regulation of six major regions of the *FOXP3* gene, and they found increased demethylation of conserved non-coding sequences in the *FOXP3* gene, which is known to be induced by TGF-β [134]. In conclusion, all these data indicate that the increased immunosuppressive activity of Tregs may play a relevant role in the pathogenesis of HIV infection and that Tregs could be potential targets for HIV immunotherapy in combination with HAART.

## 6. CD4 CTL

CD4 T cells with cytotoxic function (CD4 CTL) were initially considered to be an artifact caused by long-term in vitro cultures. However, in more recent years, they have been accepted as an important CD4+ T cell subset with cytotoxic functions produced in response to a number of viral infections or vaccinations [135]. It appears that CD4+ T cells can acquire cytotoxic activity in the presence of a combination of inflammatory signals mediated by cytokines, including IL-2 and antigen presentation [136]. These cells produce cytolytic granules, such as granzyme B, granzyme K, granzyme H, and perforin, and typical Th1 cytokines, such as IFN-γ, IL-2, and TNF [137]. CD4 T cells with a cytotoxic profile are expanded in HIV-1-infected individuals [138]. They emerge early during acute HIV infection, and their number increases in relation to the viral load [138]. By tracking T cell clonal expansion dynamics, stability, persistence, and transcriptional phenotype, Collora et al. found that HIV-1 RNA+ T cell clones had a signature as cytotoxic effector memory CD4+ T cells expressing granzyme B. These clones persisted after viral suppression, suggesting that HIV resides in these cells after suppressive antiretroviral therapy [139]. They supposed that these cells resist cell death by Bcl-2 anti-apoptotic family gene expression and are shielded from cytotoxic CD8+ T cell killing because of serpin protease inhibitor 9 (Serpin B9) expression. Immune effector cells secreting granzyme (such as cytotoxic CD8+ T cells and natural killer cells) express Serpin B9 to protect themselves from self-inflicted granule-mediated apoptosis [140]. Therefore, it may be possible that Serpin B9 expression in cytotoxic CD4+ T cells, while protecting themselves from their own granzyme B-mediated killing, can also make HIV-1-infected cells resistant to cytotoxic CD8+ T cell killing.

## 7. Naïve CD4+ T Helper Lymphocytes

Several groups have detected HIV DNA in naïve CD4+ T helper lymphocytes [141,142,143]. However, the level of HIV DNA in these cells was about 10-fold lower than in memory cells [141,142,143]. The difference in HIV DNA levels between naïve and memory cells could not be explained solely by differences in coreceptor expression but seems dependent on more efficient viral fusion in memory cells [144]. However, these cells can contribute to the HIV reservoir because they have characteristics, such as longer intermitotic half-life than memory T cells [145] and resistance to HIV expression after integration, which could create major obstacles to the eradication of HIV [144]. Moreover, infected naïve CD4+ T helper cells can give rise to infected memory T cells through differentiation and, thus, have the potential to continuously repopulate the memory subset [146].

## 8. Therapeutic Approaches to Eradicate the Viral Reservoir: “Kick and Kill” Method and Tregs Targeting

Since anti-HIV drugs act only on infected cells where the virus is in a replicative phase and are ineffective in quiescent infected cells, such as resting cells, several authors have proposed the use of viral transcription inducers in combination with HAART or immunotherapies to eradicate the viral reservoir. This approach has been named the “kick and kill” or “shock and kill” method. Most studies on reservoir cell activation have been conducted using humanized mouse models that support a level of HIV DNA persistence comparable to that of HIV-infected patients during HAART. Halper-Stromberg et al. used a combination of inducers, including vorinostat (an HDAC inhibitor), I-BET151 (a BET protein inhibitor), and αCTLA-4 (a T cell inhibitory pathway blocker), in infected humanized mice. These compounds were chosen because they were very active in inducing HIV replication in vitro without showing toxicity when administered in mice. This combination of inducers in association with broadly neutralizing antibodies (bNAbs) as a “kill” factor was able to decrease the reservoir, as measured by viral rebound, indicating that this “kick and kill” method is an effective therapeutic strategy for HIV eradication [147]. Borducchi et al. showed that therapeutic vaccination in combination with immune stimulation could be a strategy capable of eliminating the viral reservoir. They observed that therapeutic vaccination with Ad26/MVA (recombinant adenovirus serotype 26/modified vaccinia ankara) and stimulation of TLR7 (Toll-like receptor 7) decreased the level of viral DNA in lymph nodes and peripheral blood and delayed viral rebound following discontinuation of antiretroviral therapy in rhesus monkeys infected with simian immunodeficiency virus (SIV) [148]. More recently, these authors showed that V3 glycan-dependent bNAb PGT121 in combination with the TLR7 agonist vesatolimod (GS-9620) administrated during antiretroviral therapy treatment delayed viral rebound following discontinuation of antiretroviral therapy in rhesus monkeys infected with SIV [149].

A class of most effective compounds able to reverse HIV latency is the protein kinase C (PKC) modulators such as bryostatin 1 and prostratin. The bryostatin analog SUW133 is a potent HIV latent reversal agent. In addition to the reversal activity, SUW133 is capable of killing latently infected cells of mice one day after administration of the compound during antiretroviral therapy [150]. In a recent study, Kim et al. obtained an important depletion of the viral reservoir in mice using SUW133 as the “kick” factor and allogeneic natural killer cells from human peripheral blood as the “kill” factor during antiretroviral therapy. These authors did not observe a rebound of viremia despite monitoring animals for an extended time after antiretroviral therapy interruption [151]. Moreover, the virus was also undetected in mice spleens, a common hiding place for latent HIV-infected cells. This study indicates that a combined therapeutic strategy using a potent protein kinase C modulator, such as SUW133, and allogeneic human peripheral blood natural killer cells during antiretroviral therapy could successfully eliminate all infected cells.

The role of Tregs as a viral reservoir for HIV and as immunosuppressive cells has made them an object of interest for manipulation in the search for a cure for HIV. It has been shown that the depletion of Tregs activates the expression of HIV genes and induces HIV replication in CD4+ long-lasting T cells, reducing HIV-1 reservoirs during antiretroviral therapy [152,153]. Thus, Tregs depletion can act as a “kick” factor and can be associated with a “kill” factor, such as HAART. These data indicate that Tregs could be a new target for new strategies to deplete the HIV-1 reservoir and cure the infection. Currently, most scientific works have been carried out using Ontak and Daclizumab, two drugs that target Tregs through their constitutive expression of CD25. Ontak (Denileukin difitox) consists of IL-2, which binds to CD25, coupled with diphtheria toxin. Daclizumab is a monoclonal antibody against CD25, which blocks the interaction of IL-2 with IL-2R. Tregs are depleted because IL-2 is essential for Tregs development, maintenance, and function. Treatment with Ontak of African green monkeys chronically infected with SIV induced a marked reduction in Tregs and activation of CD4+ and CD8+ T-lymphocytes [154]. Similar results were observed in Rhesus macaques infected with SIV where, in addition to a marked depletion of Treg cells, there was an increase in SIV-specific T-lymphocytes [155]. Recent studies suggest that blocking CTLA-4 or PD-1 can also result in blocking the function of Tregs. Hryniewicz et al. observed that the block of CTLA-4 in macaques infected with SIV caused a decrease in viral RNA levels in lymph nodes and an increase in the effector function of both simian immunodeficiency virus-specific CD4+ and CD8+ T cells, indicating that the block of CTLA-4 could be combined with antiretroviral therapy for better control of viremia and immune response against the virus [156]. Cecchinato et al. showed that the block of CTLA-4 with anti-CTLA-4 antibodies increased T cell activation and viral replication in macaques infected with SIV, but protracted treatment decreased responsiveness to antiretroviral therapy and abrogated the ability of therapeutic T cell vaccines to decrease viral set point [157]. Other authors have found that combined targeting of PD-1 and CTLA-4 is very effective in reactivating the virus and eliminating latently infected cells [158,159,160].

Although animal studies have shown interesting results, further studies are needed to understand the benefits and risks of manipulating Tregs in HIV-infected individuals. The risks could be related to the important role of Tregs in preventing a persistent inflammatory state in HIV-infected patients that could lead to degenerative cardiovascular and neurological diseases.

## 9. Conclusions

Although the new antiretroviral therapies are very effective in controlling HIV infection, it is still not possible to completely eradicate the virus. New insights into the effects of the virus on different types of Th effectors have enabled researchers to identify the specific cellular targets of the virus and the molecular mechanisms by which the virus modifies the immune response. HIV mainly infects Th17 and Tregs subtypes, but the effects of the virus on these two cellular subtypes are profoundly different. While HIV induces a severe cytoreduction in Th17 cells, in Tregs, the virus can increase their immunosuppressive activity without causing a major reduction in their number, at least in the early stages of infection. HIV also infects Tfh cells, but their number is not particularly depleted. In contrast, their function is greatly impaired, and this prevents the production of protective antibodies against various infections and against HIV itself. Long-lived Treg cells harboring HIV insertions are important reservoirs of the virus and, unlike memory T cells harboring HIV, can escape the lytic activity of HIV-specific cytotoxic T lymphocytes due to their suppressive capacity. Due to these properties, several researchers have suggested that Treg cells are probably the most important reservoir of the virus. Therapeutic modalities are now emerging to treat HIV infection that are based on multiple approaches, such as HAART, “kick and kill” methods, and blocking Tregs development and functions. Results currently obtained in animal models, such as SIV-infected monkeys, indicate that the combination of these therapeutic methods is now the best way to attack and try to eliminate HIV from reservoirs and may, in the future, lead to a cure for people with AIDS.

## Figures and Tables

**Figure 1 ijms-25-07512-f001:**
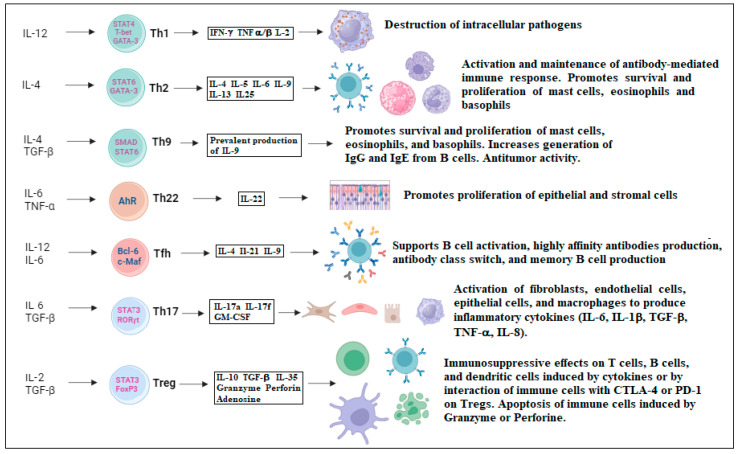
Seven subsets of Th effector cells are currently known, namely Th1, Th2, Th9, Th17, Th22, Tfh, and Tregs. Differentiation into each subtype depends on the cytokine environment present during antigen recognition. In naïve CD4+ T helper cells, these cytokines promote the production of specific transcription factors that determine differentiation into effector cells (Th1, Th2, …). The function of each Th effector cell depends on the production of specific cytokines or direct interaction with other immune cells.

## Data Availability

Not available.

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
