# Peer review of "The Complex Dysregulations of CD4 T Cell Subtypes in HIV Infection"

_ijms, 2024, doi:10.3390/ijms25147512_

Round 1

Reviewer 1 Report

Comments and Suggestions for Authors

The review concentrates on Tregs, Th17, Tfh and Th22 subsets of CD4 T cells, but, in my opinion, the review places too much emphasis on these subsets and could be greatly improved by at least more context on other subsets:

1. In high prevalence countries, it has been reported that one of the leading causes of HIV-related death is co-infection with TB, which is believed to be largely a target of Th1 immunity. Genetic defects in the IL-12/IFN-g axis are clearly associated with disseminated infections with otherwise low virulence mycobacterial infections, such as following BCG vaccination. Some reports suggest that control of TB infection may be best correlated with CXCR3+CCR6+ Th1/Th17 cells that produce both IFN-g and IL-17, rather than simply either Th1 or Th17 cells: Burel et al, doi.org/10.4049/jimmunol.1800118.

2. In my experience, Tregs are around 10% of CCR5+ CD4 T cells and Th17 are around 30% of CCR5+ CD4 T cells (using CCR6+CXCR3-CD161+ as a surrogate marker of Th17 cells, so that is an upper limit). Therefore, these cells are a minority of the target CD4 T cells for HIV, so direct infection of these cells does not seem to account for the whole story of CD4 depletion. 

At least one relatively recent report has found that Tregs are not preferentially infected with HIV proviral DNA: Dunay et al, 10.1016/j.virusres.2017.07.008. 

3. Similarly, bona fide Tfh are only found in lymph nodes and, even then, are only a small minority of memory CD4 T cells in the lymph nodes.

There are only 4 references about the Tfh subset in SIV/HIV infections, and 2 of them are from the same group reporting on dysfunction of these cells in HIV infection. However, this seems unlikely in vivo, given that all HIV+ patients, by definition, have readily detected specific antibodies. Untreated patients have extremely high titres, especially for anti-p24 antibodies. The reason these antibodies are ineffective is believed to be mainly due to HIV quasispecies that escape any neutralizing antibodies. Lymph nodes are characteristically full of germinal centres and indeed untreated patients often had hypergammaglobulinaemia, in other words the opposite of Tfh dysfunction. In other viral infections, Tfh and neutralizing antibodies often are extremely important in eventual viral clearance, as shown in COVID-19 where escape from neutralizing antibodies is still driving the epidemic. Whereas binding antibodies, but not neutralizing, antibodies are characteristic of HIV infection, along with escape from CD8 T cells. 

One other Tfh reference was a report on HIV infection of circulating Tfh, which have an unknown relationship to bona fide Tfh in lymph node germinal centres. There are many references on infection of Tfh from lymph nodes, and in particular that they have high levels of infection despite full suppression on ART: Banga et al, doi:10.1038/nm.4113.

The last reference on Tfh reported how they are infected at an early stage of differentiation, but this was in an SIV model, whereas this has been shown directly for HIV infection of human Tfh in vivo as well: Xu et al, doi: 10.3389/fimmu.2017.00376.

4. Cytotoxic effector memory CD4 T cells, including terminally differentiated granzyme B+ CD4 T cells, are not even mentioned once, but they are often a very significant subset in progressive HIV infection. There is now a consensus that such CD4 CTL are significant antiviral effector cells eg Juno et al, doi: 10.3389/fimmu.2017.00019.

Furthermore, these CD4 CTL have also recently been shown to be an important HIV reservoir cell type, particularly in the case of expanded clones of infected cells: Collora et al 

doi:10.1016/j.immuni.2022.03.004

5. Much is made of the supposed “mucosal catastrophe” of primary HIV-1 infection. This doesn’t seem to have been borne out over time, since patients that are now treated relatively early, especially with integrase inhibitor containing regimens, have limited residual T cell activation and have an apparently quite healthy prognosis.

In my experience with gut biopsies from both HIV+ patients on ART and healthy controls, only around 50%, on average, of CD4 in these samples were CD161+ memory cells, again putting an upper limit on Th17 cells - not the “80-90%” from line 298. Also, only a small minority had a CD103+ phenotype meaning that these cells are not resident memory cells and presumably just transiting through this tissue. This is consistent with a published re-evaluation that refuted the claim that most memory CD4 T cells are present in GALT: Ganusov et al; doi:10.1016/j.it.2007.08.009. The claim about CD4 T cells in GALT had been extrapolated from IgA plasmablasts, and most likely does not apply to CD4 T cells - in our experience, the number of CD4 T cells recovered in gut biopsies is surprisingly small, from either HIV+ or control donors, and was reduced about 2-fold in patients on ART, compared to healthy controls, which is similar to the situation in the circulation.

5. This raises the question of what happens to circulating naïve CD4 T cells, which are again are not discussed in the review. By the time HIV+ patients have CD4 counts < 350 cells/µl, they have largely lost their naïve CD4 T cells, from both blood and lymphoid tissue, which are difficult to recover from low nadir CD4 T cell counts after commencing ART. Dysfunction of CD4 T cell homeostasis is characteristic of untreated HIV infection, since naïve cells do not appear to be directly infected in vivo. 

One of the most characteristic dysfunctions in untreated HIV infection, even in early infection, is loss of the ability of CD4 T cells to proliferate in vitro, which has been known for a long time: Miedema et al, 0021-9738/88/12/1908/07. This may be partly ameliorated by removal of Tregs (which actually argues against Treg dysfunction) but the slow inexorable loss of CD4 T cells suggests that uninfected cells cannot expand to restore cells to refill the CD4 niche in vivo, over the long term, unless potent antiretrovirals are used, typically for years. 

There is a big gap in our knowledge about how HIV infection leads to this very slow CD4 depletion, over many years, rather than any immediate dramatic irreversible loss of any CD4 T cell subset during primary infection – some of the typical initial CD4 loss appears to result from extravasation due to significant inflammation in lymphoid tissue, and so has to be carefully interpreted, as shown by the effect of very early ART. But what happens to normal CD4 T cell homeostatic proliferation (and apoptosis) if untreated for longer still needs more study, since there don’t seem to be simple explanations.

Author Response

Comments 1. In high prevalence countries, it has been reported that one of the leading causes of HIV-related death is co-infection with TB, which is believed to be largely a target of Th1 immunity. Genetic defects in the IL-12/IFN-g axis are clearly associated with disseminated infections with otherwise low virulence mycobacterial infections, such as following BCG vaccination. Some reports suggest that control of TB infection may be best correlated with CXCR3+CCR6+ Th1/Th17 cells that produce both IFN-g and IL-17, rather than simply either Th1 or Th17 cells: Burel et al, doi.org/10.4049/jimmunol.1800118.

Response 1. In the revised manuscript we have expanded the section describing the relationship between tuberculosis and HIV by introducing the reviewer's information, especially regarding the relationship between Th1Th17 cells, HIV, and tuberculosis (pag. 6, 7, lines 217-226). In addition, an extensive description of Th1Th17 cells in HIV infection has been introduced in section 5.1. (pag. 11, 12, lines 429-437)

Comments 2. In my experience, Tregs are around 10% of CCR5+ CD4 T cells and Th17 are around 30% of CCR5+ CD4 T cells (using CCR6+CXCR3-CD161+ as a surrogate marker of Th17 cells, so that is an upper limit). Therefore, these cells are a minority of the target CD4 T cells for HIV, so direct infection of these cells does not seem to account for the whole story of CD4 depletion. At least one relatively recent report has found that Tregs are not preferentially infected with HIV proviral DNA: Dunay et al, 10.1016/j.virusres.2017.07.008.

Response 2. As reported in the introduction, all Th subtypes are potentially susceptible to HIV infection. Th17 cells express high levels of CD4, CCR5, and CXCR4 and the high levels of DNA integration of HIV in Th17 cells indicates the presence of post-entry mechanisms that enhance viral replication. Although Th17 cells does not represent the majority of CD4 T cells in blood, their important role in the defense against opportunistic infection and their severe depletion play an important role in the appearance of opportunistic infections that are the hallmark of AIDS. More references on this topic have been added in the revised manuscript.

Regarding the Treg cells and the work of Dunay' et al. these authors report that: “… no significant difference was found in total HIV DNA burden associated with Treg when compared to Tem and Tcm cells”. In our manuscript, we report that Tregs and central memory T cells are resting cells harboring HIV, but we do not specify that Tregs contain more viruses than memory Th cells. Both Tregs and central memory T cells are important reservoirs of HIV but memory T cells cannot potentially escape from the lytic activity of HIV-specific cytotoxic T lymphocytes. In contrast, long-lived Treg cells harboring HIV insertions can escape from this lytic activity, thus representing a cellular reservoir for HIV that is more difficult to eliminate. This Th subtype is not depleted (at least in the early stage of infection), but their immunosuppressive function is increased contributing to the immunosuppressed state of the HIV-infected patient. Thus, the infection of this Th subtype does not account for the whole story of CD4 depletion. In the revised paper I included the work of Dunay et al. (reference 75 and pag 10 lines 357-359). In addition, in section 5  the sentence “They (Tregs) are extremely susceptible to HIV infection showing a higher level of integration of HIV DNA” has been toned down a bit in the following sentence: “…Recent studies of well-controlled HIV-1 infection on antiretroviral therapy have shown higher frequencies of inducible, intact proviruses in Tregs compared to other CD4+ T cells, and provirus-containing Tregs have been found in lymphoid tissues at substantial frequencies (Rocco et al. 2018, PMID: 30515299). In other studies, total HIV DNA burden associated with Tregs was similar to that observed in memory T cells (Dunay,et al. 2017) …”.

Comment 3. Similarly, bona fide Tfh are only found in lymph nodes and, even then, are only a small minority of memory CD4 T cells in the lymph nodes.

There are only 4 references about the Tfh subset in SIV/HIV infections, and 2 of them are from the same group reporting on dysfunction of these cells in HIV infection. However, this seems unlikely in vivo, given that all HIV+ patients, by definition, have readily detected specific antibodies. Untreated patients have extremely high titres, especially for anti-p24 antibodies. The reason these antibodies are ineffective is believed to be mainly due to HIV quasispecies that escape any neutralizing antibodies. Lymph nodes are characteristically full of germinal centres and indeed untreated patients often had hypergammaglobulinaemia, in other words the opposite of Tfh dysfunction. In other viral infections, Tfh and neutralizing antibodies often are extremely important in eventual viral clearance, as shown in COVID-19 where escape from neutralizing antibodies is still driving the epidemic. Whereas binding antibodies, but not neutralizing, antibodies are characteristic of HIV infection, along with escape from CD8 T cells.

One other Tfh reference was a report on HIV infection of circulating Tfh, which have an unknown relationship to bona fide Tfh in lymph node germinal centres. There are many references on infection of Tfh from lymph nodes, and in particular that they have high levels of infection despite full suppression on ART: Banga et al, doi:10.1038/nm.4113.

The last reference on Tfh reported how they are infected at an early stage of differentiation, but this was in an SIV model, whereas this has been shown directly for HIV infection of human Tfh in vivo as well: Xu et al, doi: 10.3389/fimmu.2017.00376.

Response 3.  The number of Tfh cells in HIV-infected patients is not reduced and in some cases it is increased. Therefore, as reported in the manuscript, this Th subtype does not account for the whole story of CD4 depletion.

As reported in reference 45 (56 in the revised manuscript, Pallikkuth et al.), the importance of Tfh infection is that the virus can persists in these cells also after plasma virus suppression with potent antiretroviral therapy indicating that they can represent a cell compartment responsible for active and persistent virus transcription and an important reservoir of the virus contributing to HIV latency. In this contest, and according to the suggestion of the Reviewer, in the revised manuscript we reported the results of Banga et al., (doi:10.1038/nm.4113), that demonstrated that LN PD-1+ and Tfh cells from long-term-ART-treated aviremic HIV-1-infected individuals are the major source of replication-competent and infectious HIV-1 and constitute the cell compartment responsible for active and persistent virus transcription (pag. 8, 9, lines 306-312).

Regarding the hypergammaglobulinaemia observed in HIV-infected patients, a condition “in other words opposite to Tfh dysfunction”, several works consider it to be a functional disruption of B-cells in chronically infected individuals. Despite the occurrence of hypergammaglobulinemia in HIV infection, specific antibody production and in vitro B-cell differentiation responses are frequently impaired. Signs of aberrant B cell hyperactivity, including hypergammaglobulinemia, spontaneous secretion of immunoglobulins in culture and increased expression of activation markers have been observed in HIV-infected individual. However, paradoxically HIV-infected patients have an impaired humoral response, and their B cells respond abnormally when stimulated ex vivo. Chirmule et al. have observed that B cell perturbation in HIV infected patients can be caused by impaired GC T-helper cell function for B cells (PMID: 1346976). Moir et al. observed that viremia was also associated with the appearance of a subpopulation of B cells that expressed reduced levels of CD21. These B cells dramatically reduced proliferation in response to B cell stimuli and enhanced secretion of immunoglobulins when compared with normal B cells. The presence of this B cells subpopulation that fail to proliferate in response to B cell stimuli and secrete high levels of immunoglobulins may explain the hypergammaglobulinemia associated with HIV disease (DOI: 10.1073/pnas.181347898). Finally, as reported by the Reviewer, binding antibodies, but not neutralizing, antibodies are characteristic of HIV infection. All these concepts are added in the revised manuscript in the section 4, including the reference Xu et al, doi: 10.3389/fimmu.2017.00376 suggested by the Reviewer (pag. 9, lines 316-332).

Comment 4. Cytotoxic effector memory CD4 T cells, including terminally differentiated granzyme B+ CD4 T cells, are not even mentioned once, but they are often a very significant subset in progressive HIV infection. There is now a consensus that such CD4 CTL are significant antiviral effector cells eg Juno et al, doi: 10.3389/fimmu.2017.00019.

Furthermore, these CD4 CTL have also recently been shown to be an important HIV reservoir cell type, particularly in the case of expanded clones of infected cells: Collora et al doi:10.1016/j.immuni.2022.03.004.

Response 4.  In the revised manuscript we have included a section on CD4 CTLs and their role in HIV infection (pag. 15 lines 561-582)

Comment 5a. Much is made of the supposed “mucosal catastrophe” of primary HIV-1 infection. This doesn’t seem to have been borne out over time, since patients that are now treated relatively early, especially with integrase inhibitor containing regimens, have limited residual T cell activation and have an apparently quite healthy prognosis.

In my experience with gut biopsies from both HIV+ patients on ART and healthy controls, only around 50%, on average, of CD4 in these samples were CD161+ memory cells, again putting an upper limit on Th17 cells - not the “80-90%” from line 298. Also, only a small minority had a CD103+ phenotype meaning that these cells are not resident memory cells and presumably just transiting through this tissue. This is consistent with a published re-evaluation that refuted the claim that most memory CD4 T cells are present in GALT: Ganusov et al; doi:10.1016/j.it.2007.08.009. The claim about CD4 T cells in GALT had been extrapolated from IgA plasmablasts, and most likely does not apply to CD4 T cells - in our experience, the number of CD4 T cells recovered in gut biopsies is surprisingly small, from either HIV+ or control donors, and was reduced about 2-fold in patients on ART, compared to healthy controls, which is similar to the situation in the circulation.

Response 5a. On reviewing the scientific literature, I found no consensus regarding the percentage of Th17 cells in the gut, so in the revised manuscript, and also considering the Reviewer's experience, I preferred not to report precise percentages, adding only the following reference: Honda et al. doi: 10.1038/nature18848.)

Many scientific papers have been published, even very recently, on gut damage and microbial translocation in people living with HIV (Ouyang et al. https://doi.org/10.3389/fimmu.2023.1173956; Duarte et al. https://doi.org/10.1093/ofid/ofae187, and othrs). As our manuscript is a review, we have to report these works. However, according to the suggestion of the Reviewer, in the revised manuscript we also report the work of  Ganusov et al. (Ref. 13)

Comment 5b. This raises the question of what happens to circulating naïve CD4 T cells, which are again are not discussed in the review. By the time HIV+ patients have CD4 counts < 350 cells/µl, they have largely lost their naïve CD4 T cells, from both blood and lymphoid tissue, which are difficult to recover from low nadir CD4 T cell counts after commencing ART. Dysfunction of CD4 T cell homeostasis is characteristic of untreated HIV infection, since naïve cells do not appear to be directly infected in vivo.

One of the most characteristic dysfunctions in untreated HIV infection, even in early infection, is loss of the ability of CD4 T cells to proliferate in vitro, which has been known for a long time: Miedema et al, 0021-9738/88/12/1908/07. This may be partly ameliorated by removal of Tregs (which actually argues against Treg dysfunction) but the slow inexorable loss of CD4 T cells suggests that uninfected cells cannot expand to restore cells to refill the CD4 niche in vivo, over the long term, unless potent antiretrovirals are used, typically for years.

There is a big gap in our knowledge about how HIV infection leads to this very slow CD4 depletion, over many years, rather than any immediate dramatic irreversible loss of any CD4 T cell subset during primary infection – some of the typical initial CD4 loss appears to result from extravasation due to significant inflammation in lymphoid tissue, and so has to be carefully interpreted, as shown by the effect of very early ART. But what happens to normal CD4 T cell homeostatic proliferation (and apoptosis) if untreated for longer still needs more study, since there don’t seem to be simple explanations.

Response 5b. A new section on the role of naïve CD4 T cells in HIV infection has been added in the revised manuscript (pag15, lines 584-593). Regarding the mechanisms that induce CD4+ T-cell depletion and the way in which uninfected cells cannot expand to restore cells, the complexity of this topic would require another review and I do not believe it can be discussed succinctly in one section of this review.

Reviewer 2 Report

Comments and Suggestions for Authors

The authors presented a review paper on the differential susceptibility of Th subsets to HIV infection.  The review may be more appropriate as an opinion or hypothesis piece since the authors claim that Treg and Th17 are the most important subsets for targeting when developing an HIV cure.  Although some Th subsets may be more susceptible to HIV entry due to CCR5 and CXCR4 expression, all Th subsets are capable of being infected.  The main driver of productive infection is the metabolic state in "resting" versus "activated" T cells, which the authors mention.  Although the article is easy to read, the authors need to add multiple references to back up their hypothesis.  Please see specific suggestions below:

·       Run-on sentence in abstract “Although the most compromised Th subtype in HIV infection is Th17, which is implicated in the defense against opportunistic infections, HIV can induce important dysregulations in other subtypes such as Tfh cells, involved in the activation of B cells, and Tregs, a population of CD4+ T cells with immunosuppressive functions.”

·       Figure 1 under Th9 cell description, misspelled “proliferation”

·       Page 3, please add reference(s) for the claim “All these Th subtypes can mature in memory T cells. Th1, Th2, Th17, Th22, and Treg cells mature into effector memory T cells, while Tfh and Treg cells can mature into central memory T cells.”

·       Page 3, please add reference(s) for the claim “Most HIV-infected Th cells are located within lymphoid tissue of the gastrointestinal tract (gut-associated lymphoid tissue or GALT).”

·       Page 3, “HIV replication in the GALT is extremely high,” can the authors quantitatively describe how high replication is compared to blood or other compartments?

·       Page 3, please add reference(s) for the claim “In contrast, Tfh cells can be infected, but rarely they are depleted (in some cases they are increased).”

·       Table 1 should have references from primary research articles for each Th subset

·       Page 4 “Although Th1 cells express both CCR5 and CXCR4 coreceptors, they do not show high levels of integrated HIV DNA [14]” may be misleading.  In the referenced article from 2010, the study concluded that CXCR3+CCR6- T cells, which they called Th1, were not as susceptible to HIV infection as the 3 other subsets Th2, Th17, or Th1Th17 (CXCR3+CCR6+).  This Th1Th17, which isn’t usually referenced in the field, is highly permissive to infection.  A major point of this article is that some Th subsets are not major targets for HIV infection.  Please add additional references to back the claim that Th1 cells are not highly permissive to HIV infection. I’m not convinced this statement is accurate.

·       Page 4 “Th9 cell is a new subset of CD4 T cells discovered in 2008”.  Please change “new subset” to “newer subset”. 

·       Page 5, line 189 “interleukines” should be “interleukins”

·       Page 6, lines 250-251 “The number of Tfh cells in HIV-infected patients is not particularly reduced (indeed, in some cases it is increased), but their function is greatly impaired.” Please add reference(s).

·       Page 7 “are the main targets of HIV and their depletion/dysregulation is the main cause of evolution in AIDS.”  These bold claims are not backed up with evidence.  Need references.

·       Page 7, please provide references for the claims: “The drastic reduction of Th17 cells during HIV infection leads to the appearance of opportunistic infections that are the main cause of death in patients affected by AIDS. Therefore, preservation of Th17 number and functions is the primary goal of HIV treatment.”

·       Page 8, please add multiple references for “Th17 cells express high levels of CD4, CCR5, and CXCR4 and are extremely susceptible to HIV infection showing higher level of integration of HIV DNA compared to Th1 and Th2 cells.”  If primary reference is [14], please add secondary reference.

·       Page 8 “In addition, at an early stage of infection, HIV is able to inhibit in Th17 cells the autophagy induced by mTORc1 (mammalian Target Of Rapamycin Complex 1) [66]. mTORc1-mediated autophagy is a mechanism to eliminate infected cells including cells infected by HIV.”  mTORC1 does not induce autophagy, it inhibits autophagy.  mTORC1 is beneficial for HIV replication.

·       Page 8, line 366 “As FoxP3, CTLA-4 is critical”.  Change “As FoxP3” to “Like FoxP3” or “Similar to FoxP3”

·       Page 10, line 447, “Foxp3 is capable to inhibits RORγt”, “inhibits” should not be plural

·       Page 10, line 470, remove comma within the section title

·       Page 10, line 470, remove “new”.  The “kick and kill” strategy has been attempted by several groups for over a decade.

·       Page 11, line 508 “The most important HIV reservoirs are the memory T cells, as well as Treg cells.”  This is an opinion.  Please explain what “most important” means.

·       Page 12, at the end of the Treg targeting section, please explain the caveats to this approach.  Removing Treg populations will lead to autoimmunities.  Also, the majority of latency reversing agents are toxic with side effects to bystander cells.  Although these approaches have been intensely studied for more than a decade, nothing fruitful has come from the “kick and kill” approach.

Author Response

Comment 1. Run-on sentence in abstract “Although the most compromised Th subtype in HIV infection is Th17, which is implicated in the defense against opportunistic infections, HIV can induce important dysregulations in other subtypes such as Tfh cells, involved in the activation of B cells, and Tregs, a population of CD4+ T cells with immunosuppressive functions.”

Response 1. We have shortened this sentence.

Comment 2.  Figure 1 under Th9 cell description, misspelled “proliferation”.

Response 2. The word has been corrected

Comment 3.  Page 3, please add reference(s) for the claim “All these Th subtypes can mature in memory T cells. Th1, Th2, Th17, Th22, and Treg cells mature into effector memory T cells, while Tfh and Treg cells can mature into central memory T cells.”

Response 3. References have been added.

Comment 4.  Page 3, please add reference(s) for the claim “Most HIV-infected Th cells are located within lymphoid tissue of the gastrointestinal tract (gut-associated lymphoid tissue or GALT).”

Response 4. References have been added.

Comment 5. Page 3, “HIV replication in the GALT is extremely high,” can the authors quantitatively describe how high replication is compared to blood or other compartments?

Response 5. In the revised manuscript this sentence has been changed considering that not all the authors agree with this statement and the percentages reported in the literature of HIV-infected cells in GALT are highly variable. We have therefore modified this section of the manuscript as follows, also reporting data from authors who disagree with this statement: “… Most HIV-infected Th cells are located within lymphoid tissue of the gastrointestinal tract (gut-associated lymphoid tissue or GALT) and within the lamina propria of the gut. Th cells residing in these areas are primarily infected and depleted. The intestinal barrier, that must limit the entry of microbes from the gut into the systemic circulation, can be destroyed by HIV infection and the increased intestinal permeability leads to translocation of microbial products from the gut to the blood, contributing to increased immune activation, risk of non-AIDS-related comorbidities and mortality in people living with HIV (Ouyang, J.; Yan, J.; Zhou, X.; Isnard, S.; Harypursat, V.; Cui, H.; Routy, J.P.; Chen, Y. Relevance of biomarkers indicating gut damage and mi-crobial translocation in people living with HIV. Front. Immunol. 2023, 14, 1173956.). However, reviewing data from different mammalian species, other authors have found that only 5%–20% of all lymphocytes reside in the gut, suggesting that spleen and lymph nodes, rather than the gut, are the largest immune compartments in mammals (Ganusov, V.V.; De Boer, R.J. Do most lymphocytes in humans really reside in the gut? Trends Immunol. 2007, 28, 514-8.) …” (pag. 3, 4, lines 94-104)

Comment 6. In Page 3, please add reference(s) for the claim “In contrast, Tfh cells can be infected, but rarely they are depleted (in some cases they are increased).”

Response 6. References have been added.

Comment 7. Table 1 should have references from primary research articles for each Th subset.

Response 7. References have been added.

Comment 8. Page 4 “Although Th1 cells express both CCR5 and CXCR4 coreceptors, they do not show high levels of integrated HIV DNA [14]” may be misleading.  In the referenced article from 2010, the study concluded that CXCR3+CCR6- T cells, which they called Th1, were not as susceptible to HIV infection as the 3 other subsets Th2, Th17, or Th1Th17 (CXCR3+CCR6+).  This Th1Th17, which isn’t usually referenced in the field, is highly permissive to infection.  A major point of this article is that some Th subsets are not major targets for HIV infection.  Please add additional references to back the claim that Th1 cells are not highly permissive to HIV infection. I’m not convinced this statement is accurate.

Response 8. In reference 14 (reference 19 in the revised manuscript) authors reported that “…CXCR3+CCR6− T cells expressed CCR5 and CXCR4 but were relatively resistant to R5 and X4 HIV in vitro (Abstract)”. Therefore, Th1 cells express both HIV co-receptors but are relatively resistant to HIV infection. In Results and Discussion, the authors of reference 14 report that “… CCR6+ T cells (such as Th17 and Th1Th17) compared with CCR6− T cells (such as Th1) harbored higher levels of integrated HIV DNA”.  Therefore, cells expressing CCR6, (Th17 and in Th1Th17 cells) hve higher levels of integrated HIV DNA compared to CCR6- T cells (Th1). To avoid confusion, in the revised manuscript we have reported the exact sentences of reference 14 (pag. 5, lines 163-172). In addition, to support this assertion, in the revised manuscript we added the following reference: “Singh, A.; Vajpayee, M.; Ali, S.A.; Chauhan, N.K. Cellular interplay among Th17, Th1, and Treg cells in HIV-1 subtype "C" infection. J. Med. Virol. 2014, 86, 372-84”. Finally, in the revised manuscript, a section on Th1Th17 (CXCR3+CCR6+) cells has been added (pag. 11-12, lines429-437).  

Comment 9. Page 4 “Th9 cell is a new subset of CD4 T cells discovered in 2008”.  Please change “new subset” to “newer subset”.

Response 9. We changed “new subset” to “newer subset”.

Comment 10. Page 5, line 189 “interleukines” should be “interleukins”.

Response 10. “interleukines” has been changed in “interleukins”.

Comment 11. Page 6, lines 250-251 “The number of Tfh cells in HIV-infected patients is not particularly reduced (indeed, in some cases it is increased), but their function is greatly impaired.” Please add reference(s).

Response 11. A reference has been added.

Comment 12. Page 7 “are the main targets of HIV and their depletion/dysregulation is the main cause of evolution in AIDS.”  These bold claims are not backed up with evidence.  Need references.

Response 12. In the revised manuscript this sentence has been changed as follow: “Th17 cells and Treg cells (regulatory T cells or Treg cells) are two of the main targets of HIV, and their depletion (Th17) and dysregulation (Tregs) contribute to the evolution in AIDS” (pag. 9, 10, lines 348-350). References have been added.

Comment 13.  Page 7, please provide references for the claims: “The drastic reduction of Th17 cells during HIV infection leads to the appearance of opportunistic infections that are the main cause of death in patients affected by AIDS. Therefore, preservation of Th17 number and functions is the primary goal of HIV treatment.”

Response 13. References have been added.

Comment 14. Page 8, please add multiple references for “Th17 cells express high levels of CD4, CCR5, and CXCR4 and are extremely susceptible to HIV infection showing higher level of integration of HIV DNA compared to Th1 and Th2 cells.”  If primary reference is [14], please add secondary reference.

Response 14. References have been added

Comment 15.  Page 8 “In addition, at an early stage of infection, HIV is able to inhibit in Th17 cells the autophagy induced by mTORc1 (mammalian Target Of Rapamycin Complex 1) [66]. mTORc1-mediated autophagy is a mechanism to eliminate infected cells including cells infected by HIV.”  mTORC1 does not induce autophagy, it inhibits autophagy.  mTORC1 is beneficial for HIV replication.

Response 15. This sentence is wrong and in the revised manuscript has been removed.

Comment 16. Page 8, line 366 “As FoxP3, CTLA-4 is critical”.  Change “As FoxP3” to “Like FoxP3” or “Similar to FoxP3”.

Response 16. “As FoxP3” has been changed to “Like FoxP3”.

Comment 17. Page 10, line 447, “Foxp3 is capable to inhibits RORγt”, “inhibits” should not be plural.

Response 17.  “inhibits” has be changed to “inhibit”.

Comment 18. Page 10, line 470, remove comma within the section title.

Response 18. Comma has been removed. 

Comment 19. Page 10, line 470, remove “new”.  The “kick and kill” strategy has been attempted by several groups for over a decade.

Response 19. We have removed “new”.

Comment 20. Page 11, line 508 “The most important HIV reservoirs are the memory T cells, as well as Treg cells.”  This is an opinion.  Please explain what “most important” means.

Response 20. Considering the context of this sentence, we preferred to replace it with the following sentence: “The role of Tregs as a viral reservoir for HIV and as immunosuppressive cells has made them an object of interest for manipulation in the search for a cure for HIV” pag.16, lines 635-636). In reference 125 is reported that “… this reservoir is mainly composed of cells with a memory phenotype …”. The role of Tregs as reservoirs is discussed in the section 5.2.

Comment 21. Page 12, at the end of the Treg targeting section, please explain the caveats to this approach.  Removing Treg populations will lead to autoimmunities.  Also, the majority of latency reversing agents are toxic with side effects to bystander cells.  Although these approaches have been intensely studied for more than a decade, nothing fruitful has come from the “kick and kill” approach.

Response 21. As reported at the end of the section “Therapeutic approach”, “ (pag. 17, lines 662-666) …. Although animal studies have shown interesting results, further studies are needed to understand the benefits and risks of manipulating Tregs in HIV-infected individuals. The risks could be related to the important role of Tregs in preventing a persistent inflammatory state in HIV-infected patients that could lead to degenerative cardiovascular and neurological diseases ….” Although the 'Kich and Kill' method has currently produced controversial results, we believe that this therapeutic approach can be improved in the future (for example, see the recent article of Klinnert et al. published on Gene Ther. 2024 Mar;31(3-4):74-84. doi: 10.1038/s41434-023-00413-1.) or used in combination with multiple approach with the aim of removing the virus from the cellular reservoirs.

Round 2

Reviewer 1 Report

Comments and Suggestions for Authors

The review is now more comprehensive and provides a better context for the authors’ arguments.

Just a couple of minor comments, including 2 that I forgot to put in before. 

I think that Th0 cells is an old term, that wasn’t meant to be naive cells, but I thought it referred to uncommitted memory cells just before they differentiated down the more Th1 or Th2 specialized pathway. But I couldn’t find a recent review that mentions Th0 anymore. They may have been Th9/17/22 cells, Tfh or CD4 CTL before those newer subsets were described.

I have now noticed that the Figure shows granzymes in Tregs. We measure granzymes and perforin all the time in CD4 T cells in blood and they are not present in circulating Tregs. Maybe they arise after activation, but we have never looked at this specifically.

Note about HIV tropism: not everyone seems to have HIV-1 quasispecies that have switched significantly to X4.

I still think there is generally too much emphasis on gut depletion, but that is my personal opinion based on our own observations. Prof Douek recently stated that a very large majority of CD4 T lymphocytes are found in “lymphatic tissues” - he included GALT as lymphoid tissue. Then he argued that after HIV infection most CD4 T cells from GALT are depleted so that the large majority of HIV infected reservoir cells are in other lymphoid tissues, esp lymph nodes and spleen. I think this is correct, but we found that in absolute terms, the decrease in the number of CD4 T cells in gut biopsies was not as dramatic as they keep saying. Also, for example, he never seems to mention skin, which is actually the largest organ and does contain CD4 T cells, but CCR4+ CD4 T cells likely to represent skin homing CD4 T cells and they can be quite a significant subset in blood, especially CCR4+ Tregs. Dermal pathological conditions are common in untreated HIV infection. Likewise, adipose tissue, where Tregs have been shown to be prominent in mice. 

Line 297. Should be “lymph nodes”

As I understand it, the latency reversal therapy approach has not shown beneficial effects in relatively recent clinical trials

Most of the encouraging work quoted in the review about checkpoint inhibitors is from the SIV model. 

I became aware recently about a Phase 1b study presented at CROI 2024 (Abstract 106, Krishnan et al) which reported that PD-1 blockade during an analytical treatment interruption led to viral suppression in a subset of the participants, with a trend toward changes in effector CD4 and CD8 T cells, possibly indicating immune control. 

So the overall conclusion about targeting PD-1+ Tregs is extremely topical.

Comments on the Quality of English Language

Line 297 should be "lymph nodes"

Author Response

Comment 1: I think that Th0 cells is an old term, that wasn’t meant to be naive cells, but I thought it referred to uncommitted memory cells just before they differentiated down the more Th1 or Th2 specialized pathway. But I couldn’t find a recent review that mentions Th0 anymore. They may have been Th9/17/22 cells, Tfh or CD4 CTL before those newer subsets were described.

Response 1: To avoid confusion, and according to the Reviewer, in the revised manuscript I have removed the abbreviation “Th0” and replaced it with “naïve CD4+ T helper cells”.

Comment 2: I have now noticed that the Figure shows granzymes in Tregs. We measure granzymes and perforin all the time in CD4 T cells in blood and they are not present in circulating Tregs. Maybe they arise after activation, but we have never looked at this specifically.

Response 2: Since this manuscript is a Review, we have to report all the most important and documented information on the subject. There are several papers on the ability of Treg cells to produce granzyme and perforin. In the revised manuscript, we have introduced four of them in the section “References”. These papers are: 1) Sun et al.  Granzyme B-expressing Treg cells are enriched in colorectal cancer and present the potential to eliminate autologous T conventional cells. Immunol. Lett. 2020, 217, 7-14. 2) Loebbermann et al.  Regulatory T cells expressing granzyme B play a critical role in controlling lung inflammation during acute viral infection. Mucosal Immunol. 2012 Mar;5(2):161-72.  3) Velaga et al.  Granzyme A Is Required for Regulatory T-Cell Mediated Prevention of Gastrointestinal Graft-versus-Host Disease. PLoS One 2015, 10, e0124927.  4) Cao et al. Granzyme B and perforin are important for regulatory T cell-mediated suppression of tumor clearance. Immunity 2007, 27, 635-46. Many other works on this subject can be found in the literature. Considering the reviewer's experience, granzyme production could take place following the activation of the Tregs to perform their functions.

Comment 3: Note about HIV tropism: not everyone seems to have HIV-1 quasispecies that have switched significantly to X4. I still think there is generally too much emphasis on gut depletion, but that is my personal opinion based on our own observations. Prof Douek recently stated that a very large majority of CD4 T lymphocytes are found in “lymphatic tissues” - he included GALT as lymphoid tissue. Then he argued that after HIV infection most CD4 T cells from GALT are depleted so that the large majority of HIV infected reservoir cells are in other lymphoid tissues, esp lymph nodes and spleen. I think this is correct, but we found that in absolute terms, the decrease in the number of CD4 T cells in gut biopsies was not as dramatic as they keep saying. Also, for example, he never seems to mention skin, which is actually the largest organ and does contain CD4 T cells, but CCR4+ CD4 T cells likely to represent skin homing CD4 T cells and they can be quite a significant subset in blood, especially CCR4+ Tregs. Dermal pathological conditions are common in untreated HIV infection. Likewise, adipose tissue, where Tregs have been shown to be prominent in mice.

Response 3. We fully respect the opinion and experience of the Reviewer; he has given us important information to improve this Review. In the revised manuscript we introduced a new reference on the topic GALT and HIV infection published in 2023 on IJMS: " Moretti, S.; Schietroma, I.; Sberna, G.; Maggiorella, M.T.; Sernicola, L.; Farcomeni, S.; Giovanetti, M.; Ciccozzi, M.; Borsetti, A. HIV-1-Host Interaction in Gut-Associated Lymphoid Tissue (GALT): Effects on Local Environment and Comorbidities. Int. J. Mol. Sci. 2023, 24, 12193”. Although we do not know Prof Douek's first name, there are three references in this work where the name Douek is mentioned.

Comment 4: Line 297. Should be “lymph nodes”

Response 4: We have corrected “limpho nodes” in “limph nodes”.